# Identification of Priority Forest Conservation Areas for Critically Endangered Lemur Species of Madagascar

Virginia E. García Millán [1,*], David Rodríguez-Rodríguez [1], Amanda Martin Oncina [1], Aristide Andrianarimisa [2,3], Lalatiana O. Randriamiharisoa [4], Gabriel Martorell-Guerrero [1], Antonio Bóveda [4] and Dania Abdul Malak [1]

[1] European Topic Centre on Spatial Analysis and Synthesis, University of Malaga, Ada Byron Research Building, C/Arquitecto Francisco Peñalosa, 18, 29010 Malaga, Spain
[2] Department Zoology and Animal Biodiversity, Faculty of Science, University of Antananarivo, P.O. Box 906, Antananarivo 101, Madagascar
[3] Wildlife Conservation Society, Madagascar Program, Villa Ifanomezantsoa, Soavimbahoaka, P.O. Box 8500, Antananarivo 101, Madagascar
[4] Madagascar National Parks, Lot AI C 10 Ambatobe, BP 1424, Antananarivo 103, Madagascar
* Correspondence: virginia.garcia@uma.es

**Abstract:** Forests have extraordinary importance for the conservation of endemic species in Madagascar. However, they are disappearing fast due to a number of pressures, notably unsustainable agricultural practices leading to aggravated status of biodiversity. Here, we used a number of ecological and spatial criteria to identify and prioritise unprotected forest areas for the conservation of the eight critically endangered species of lemur belonging to the *Lemuridae* family in Madagascar. By combining spatial information layers on the distribution areas of the studied lemurs, forest extension and conservation status, and potential human impacts (such as roads, human settlements and agriculture lands), it was possible to identify the most appropriate sites for the expansion of the conservation areas of critically endangered lemur species. Seven new sites, totalling over 33,000 ha, were identified as priority sites for the protection of those species. All of them were adjacent to or inside (just one site) existing protected areas (PAs), which likely makes their protection both feasible and socioeconomically efficient by enlarging those PAs. Legally protecting these sites would not only take Madagascar one little step ahead for meeting oncoming global biodiversity targets for 2030 but could also make a substantial contribution to the mid-term survival of the studied lemur species.

**Keywords:** biodiversity hotspot; pressure; protected area; fragmentation; habitat

## 1. Introduction

Madagascar and the surrounding Western Indian Ocean islands are considered one of the world's top three biodiversity hotspots, where exceptional richness of endemic species coexists with high levels of habitat loss that threatens their long-term survival [1]. Five of the world's 238 priority ecoregions for conservation are located in Madagascar, and four of those biomes were already considered critically endangered in the early 2000s [2]. Madagascar's endemicity rates are astonishing, with 100% of its amphibian species, 96% of its vascular plant species, 90% of its reptile species and 100% of its 104 species and subspecies of lemurs to be found only in that country [3]. Such endemicity rates derive from the islands' long-lasting geological isolation after being detached from Gondwana some 160 million years ago and from current India roughly 90 million years ago [4].

Most endemic species in Madagascar, including all but one lemur species (*H. alaotrensis*), are forest species [5–8]. They depend on well-preserved forests for their survival. However, forests are being degraded rapidly in Madagascar to provide agricultural land, pastures for cattle and heating and building material for impoverished local populations [7,9,10]. Recent studies have estimated that the island lost up to half of its forest cover in the second half of the 20th century alone [11], with the remaining forests covering around 15% of the island [8]. To

combat such a trend, the Malagasy government embarked on a process of rapid expansion of its terrestrial protected area (PA) system, which rose from 2.9% of the country's terrestrial area in 2003 to over 12% in 2016 [7]. Notwithstanding the huge effort required to put aside such an amount of territory for biodiversity conservation by one of the world's poorest countries [12], a number of challenges remain, both inside and outside PAs. Among them, scarce financial resources for management, weak law enforcement, poverty, corruption, poaching, illegal logging, charcoal production, mining and political instability make the long-term conservation of biodiversity challenging inside PAs [7,8,13,14]. Outside PAs, the situation is more critical, with a range of uncontrolled pressures affecting the remaining natural habitats, notably through slash and burning [6]. Poverty and uncertainty in land ownership make most rural Malagasy resort to quick and cheap slash and burn of forests for cultivation under a 'first use, continuous occupancy' informal ownership scheme [15]. Such a traditional land occupation system grants land use and ownership to year-long first cultivators under an unsustainable cycle that provides no incentives for conserving forests or soils, which eventually degrade and lose their agricultural potential [16]. As a result, just a small proportion of the country's forest areas remains outside PAs [8].

Lemurids are an evolutionarily distinct group of primates that evolved in isolation in Madagascar and a few other surrounding islands over approximately 100 million years [5,9]. As a result of intense deforestation and hunting, at least six genera and fourteen large species of lemurs have become extinct since the arrival of humans on the island some two thousand years ago, and most other species have seen drastic population declines [5,9,17]. Currently, 101 species remain in Madagascar, although 94% of them are threatened with extinction [6,17]. The evolutionary importance of primate conservation in Madagascar is the world's greatest; five primate families, representing 20% of the world's primate species, are found nowhere else [6]. Currently, eight lemur species from the *Lemuridae* family that inhabit Madagascar are facing an extremely high risk of extinction in the wild and are, thus, internationally considered as critically endangered [18].

Most existing lemur populations remain in relative safety inside PAs, although important pressures such as deforestation or poaching remain [6,17,19]. International targets aim to substantially increase the coverage of PAs and other effective conservation measures by 2030 to cover at least 30% of important areas for biodiversity and ensure their effectiveness [3]. Expanding the protection of remaining forest habitats and managing existing PAs more effectively are considered essential to lemur conservation [6]. However, whereas intense land demand for farming makes substantial land protection enlargement unlikely, chronic financial and managerial constraints in PAs could make it largely ineffective, posing a substantial challenge to the survival of those species [7,8,13].

This collaborative study includes some key biodiversity research organisations and PA-managing organisations in Madagascar. It seeks to identify viable, unprotected natural forest areas and prioritise them for targeted and feasible enlarged habitat protection for the conservation of some critically endangered lemur species.

## 2. Materials and Methods

### 2.1. Lemur Species under Study

Eight species of the *Lemuridae* family were identified as critically endangered globally [18]. Table 1 synthesises the ecological and conservation features of the studied lemur species.

**Table 1.** Ecological characteristics of the eight critically endangered species of the *Lemuridae* family of Madagascar.

| Species | Distribution Area (ha) | Habitat | Forest in Distribution Area (%) | Distribution Area Protected (%) | Minimal Habitat Size (ha) | Number of Individuals [1] | Reference |
|---|---|---|---|---|---|---|---|
| Golden Bamboo Lemur (*Hapalemur aureus*) | 239,278 | Forest; Wetland | 77.65 | 87 | 10 to 30 | 3 to 4 | [20] |
| Black-and-White Ruffed Lemur (*Varecia variegata*) | 2,071,712 | Forest | 73.03 | 57 | 15 to 197 | 8 to 16 | [21–23] |
| Blue-Eyed Black Lemur (*Eulemur flavifrons*) | 241,708 | Forest | 22.76 | 17 | 6.6 to 8.5 | 4 to 11 | [24,25] |
| Greater Bamboo Lemur (*Prolemur simus*) | 170,904 | Forest | 78.71 | 79 | 40 to 60 | 4 to 7 | [26,27] |
| Red Ruffed Lemur (*Varecia rubra*) | 526,763 | Forest | 71.63 | 78 | 23 to 58 | 5 to 31 | [28] |
| Mongoose Lemur (*Eulemur mongoz*) | 636,149 | Forest | 28.20 | 63 | 2.8 to 2.9 | 3 to 4 | [29–31] |
| Alaotra Reed Lemur (*Hapalemur alaotrensis*) | 21,633 | Wetland | 4.71 | 100 | 1 to 2 | 2 to 9 | [32] |
| White-Collared Lemur (*Eulemur cinereiceps*) | 181,702 | Forest | 75.41 | 86 | 7 to 20 | 3 to 12 | [33] |

[1]. In minimal habitat patches.

### 2.2. Site Selection Criteria

New conservation areas must ideally be large and non-fragmented forest patches at some distance from (i.e., not adjacent to) human pressures to facilitate the effective conservation of the critically endangered lemur species [34,35]. Thus, the quality of potential new conservation areas for lemurs was based on the quality of their forests from a structural point of view, considering factors such as total forest area, average patch size, number of forest patches, spatial aggregation and/or dispersion of forest patches, compactness, number of forest edges, number of forest perforations and inter-patch connectivity. A two-step spatial analysis was used (Table 2). First, a combination of several geospatial information layers was performed to identify three potential new conservation areas per species, considering the presence of forests and human pressures. Second, the quality of forests within the three proposed areas was evaluated in terms of forest size and spatial pattern to select the best option of the three. Table 2 summarises the criteria used to identify potential new PAs for the studied critically endangered lemur species.

**Table 2.** Methodological flow and criteria used for the selection of candidate sites.

| Selection Criteria | |
|---|---|
| **Step 1** | |
| 1.1 | The areas must be within the IUCN distribution areas of the studied lemur species. If more than an area of distribution exists, the largest one was selected |
| 1.2. | The area must contain forests, as the studied lemur species only or mostly live in this ecosystem. |
| 1.3. | The areas must be out of protected areas, as the study tries to propose new protected areas or the enlargement of existing ones. |
| 1.4. | The areas must be at some distance from human pressures. The following sources of impacts were considered, as well as a buffer area of influence: |
| | i. Human settlements—distance of 1 km. |
| | ii. Croplands—distance of 100 m to agriculture patches that are larger than 0.5 ha. |
| | iii. Roads—distance of 500 m on both sides for major and minor roads, 100 m on both sides of very small roads and car-free paths. |
| 1.5. | From all the potential candidates, the three largest forest patches were proposed as potential new conservation areas. |
| **Step 2** | |
| 2.1 | The quality of forest patches was quantified according to: |
| | i. Patch size—the area with larger forest patches is preferable. |
| | ii. Connectivity—the areas where the forest patches are connected are better. |
| | iii. Fragmentation—the less fragmented forest patches are more desirable. |
| 2.2 | Overlap of different species distribution areas. |
| 2.3. | Overlap of distinct conservation candidate sites. |
| 2.4. | Adjacency to existing protected areas. |

In step 2, a simple score system was applied to rank the three proposed new conservation areas, per each of the studied lemur species, to select the final candidate sites. For criteria 1 to 2.1, each of the spatial criteria was scored from 3 to 1, 3 being the highest score. Then, an extra point was given if the site harbours more than one of the critically endangered lemur species (criterion 2.2). Furthermore, the site received extra points if the area hosts more than one of the studied lemur species (one extra point per species; criterion 2.3) or if the site is adjacent to an existing PA (criterion 2.4). At the end, all the points were summed up for the valuation of sites.

### 2.3. Spatial Data and Spatial Analysis

Spatial data were used to search for potential new areas of conservation for the eight species according to the ecological criteria listed in Table 2. Each of the criteria was addressed with an information layer in a geographic information system (GIS).

### 2.3.1. Step 1—Selecting Potential New Conservation Areas

The distribution areas of each of the studied lemur species were explored in ArcGIS v.10.8. using the data provided by the International Union on Conservation of Nature [18]. The data were provided in polygon shapefile format. If the distribution area of a given lemur species spread into more than one location, the largest area (polygon) was selected. Information about the extension of forests in Madagascar was extracted from the land cover map of Madagascar [36], which was generated based on Sentinel-2 MSI images (10 m pixel resolution, raster format).

The location of protected areas in Madagascar was retrieved from the World Database on Protected Areas (WDPA, September 2021 version) [37], also in polygon shapefile format. Human impacts were depicted from the land cover map of Madagascar [36] in the case of croplands, while roads and human settlements were extracted from Open Street Map [38] in linear and polygon shapefile format, respectively.

The data were first pre-processed. Firstly, the land cover maps (which were used to represent croplands and forests) were re-projected to the African Conic Albers reference system to maintain equal areas. Then, a distance buffer was calculated for the human impact layers (roads, croplands and human settlements) to include the immediate area of influence of humans over potential lemur areas, resulting in the distance values reported in Table 2.

All previously mentioned layers were intersected to select the forest areas within the lemurs' distribution areas, outside protected areas and at some distance from human pressures (Figure 1).

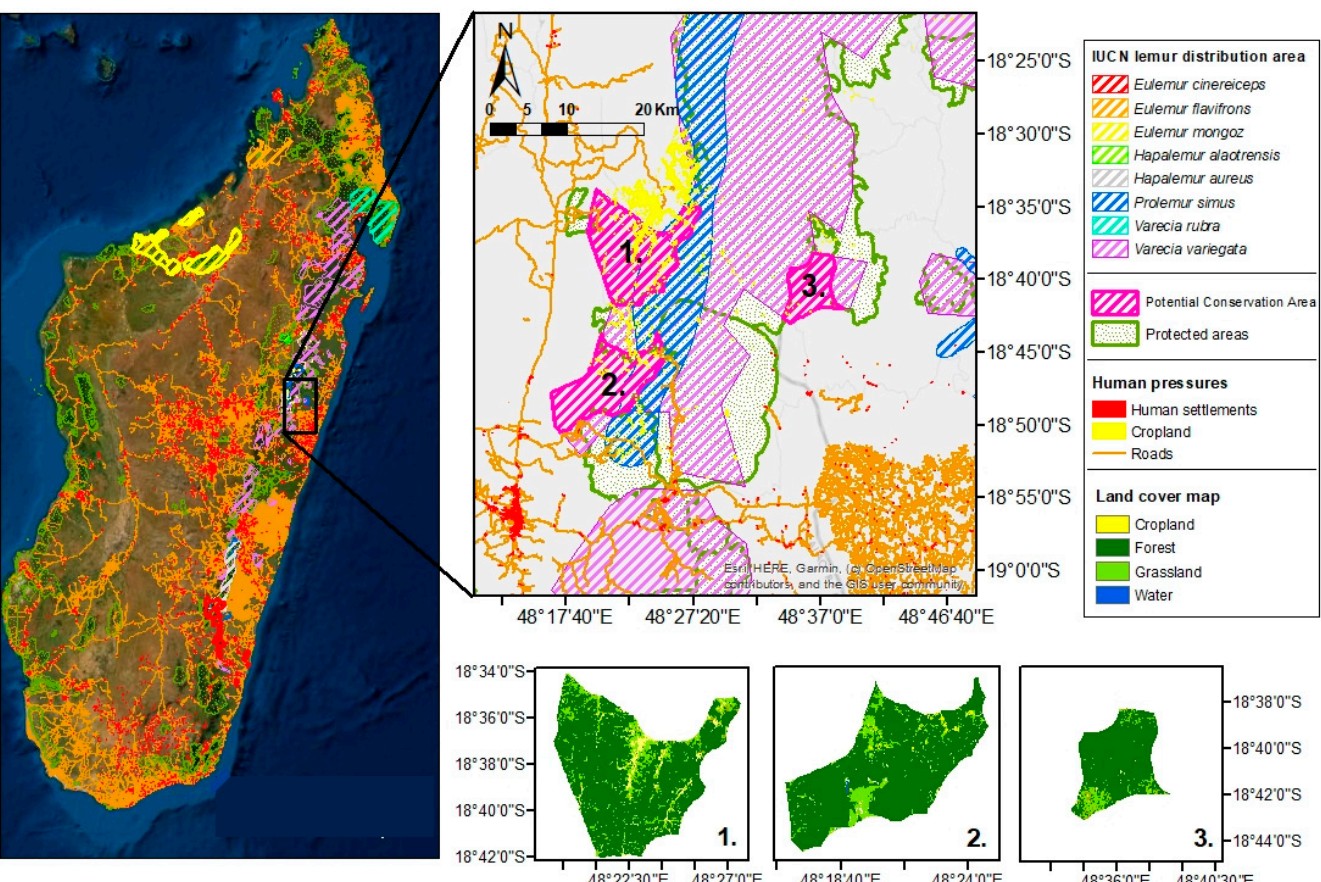

**Figure 1.** Three proposed new conservation areas extracted from the first spatial analysis. *Varecia variegata* is shown as a case example. *Source*: Esri, Maxar, GeoEye, Earthstar Geographics, CNES/Airbus DS, USDA, USGS, AeroGRID, IGN, and the GIS User Community.

Then, those forest patches were classified by size range using the tool "Accounting" in the GUIDOS toolbox [39]. From this, the three largest forest patches were selected for each

of the eight studied lemur species. A boundary around each of these three forest patches (per lemur species) was defined to define a potential new PA, which may include other land cover classes beyond forests (Figure 1).

### 2.3.2. Step 2—Evaluating the Quality of the Three Largest Forest Patches

In a second step, the three potential new conservation areas selected in step 1 were evaluated to find the most suitable area of conservation for each of the studied lemur species. The evaluation was made according to forest quality. Forest quality was defined in terms of patch size, connectivity and fragmentation (Figure 2). The GUIDOS toolbox [39] was used to generate new layers of information that represent these variables. Each of these new spatial variables was ranked between 1 and 3, 3 being the best option and 1 the least suitable option.

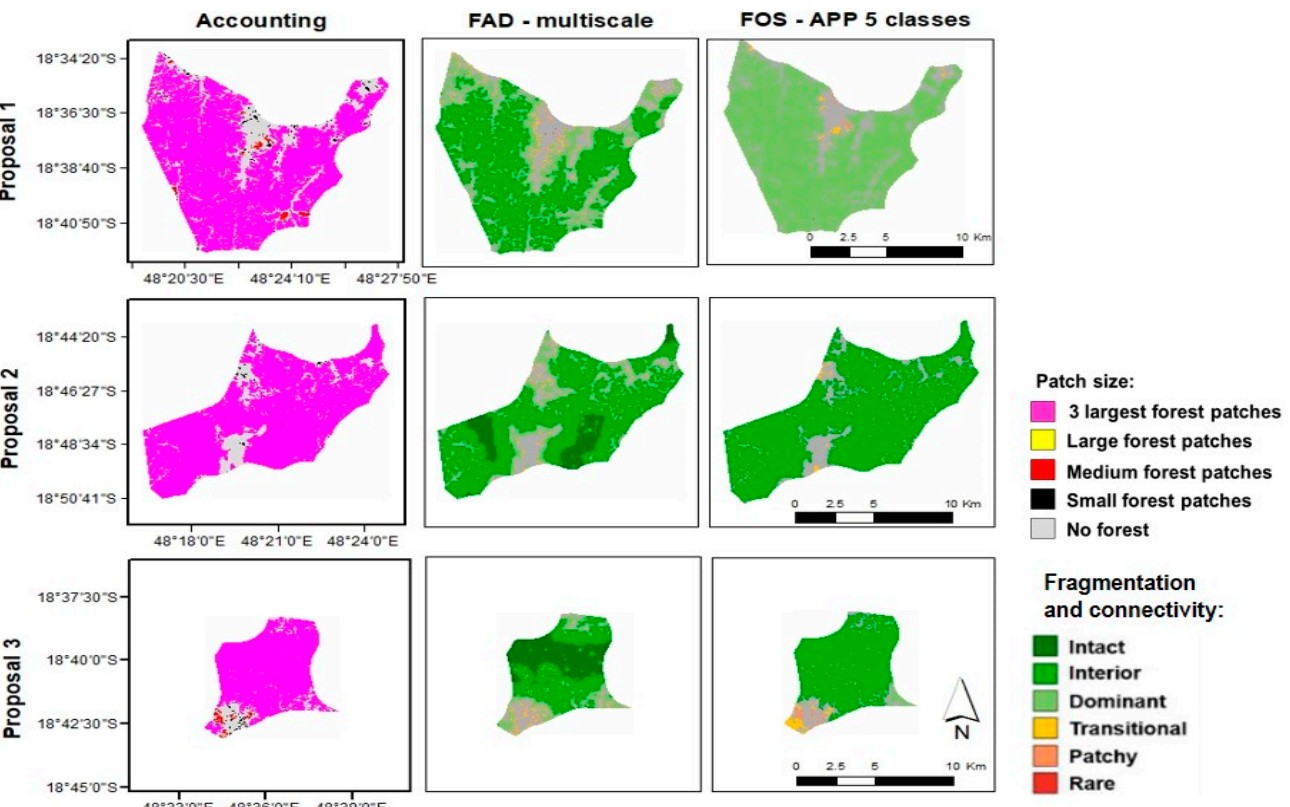

**Figure 2.** Spatial analysis results for the best new conservation areas. *Varecia variegata* is shown as a case example.

- Patch size. Forest patches were classified into 5 classes based on a size range defined by the user using the GUIDOS "Accounting tool". For the final ranking, the following variables were considered: (a) largest size range; (b) % of forests of the largest size range class.
- Forest area density (FAD) [40]. This variable accounts for forest integrity, defined as the number, shape and area of perforations inside forests (inner fragmentation) and the spatial distribution of forest patches separated by non-forest lands (connectivity). The tool quantifies the integrity of forests from "intact" to "highly fragmented" at different spatial scales and integrates the results into an average value, classifying forest integrity into: intact, interior, dominant, transitional, patchy and rare. For the selection of the best candidate for a new conservation area, the percentages of the classes "intact" and "interior" were accounted for.
- Fixed observation scale (FOS) [41]. The per-pixel forest density values calculated in the FAD analysis were averaged for each forest patch (average-per-patch (APP) value),

giving a general overview of the quality of a forest patch. For ranking the final new conservation area candidate, only percentages of forests in the classes "intact" and "dominant" were taken into account.

Three additional factors were considered as extra points for the final ranking of proposed sites: adjacency to existing protected areas, whether they covered the distribution area of another studied lemur species and whether they overlapped with proposed conservation areas for another of the studied lemur species. The connection of potential sites to existing PAs and the overlapping habitat of several lemur species were evaluated in ArcGIS 10.8.

## 3. Results

The size of the new proposed conservation sites varied in range. Some of them were thousands of hectares large (*E. cinereceips*, *E. flavifrons*, *P. simus*, *V. rubra* and *V. variegata*), while others were around hundreds of hectares (*E. mongoz*, *H. alaotrensis* and *H. aureus*). In general, the percentage of forest within the proposed areas was around 50–100%. In the case of the final candidate areas, this percentage was even larger, around 70–100%, with the exception of the area for *H. alaotrensis* (15%). Table 3 summarises all the quantitative structural spatial variables observed for forest patches sorted for each of the eight studied lemur species.

The new candidate sites amounted to 33,587 ha, of which 33,052 ha do not currently have a legal protection category. In terms of their potential contribution to lemur conservation, the number of protected populations and individuals would increase substantially (Table 4).

The spatial analysis revealed that all the selected sites are adjacent to existing PAs (Figure 3; Table 5).

**Table 3.** Description and scoring of the forest spatial context: extension, connectivity and fragmentation within the distribution area of the eight selected critically endangered lemur species in Madagascar. The highest score is shown in bold.

| | Site | Size (ha) | Score | Forest Area (%) | Score | % of Largest Forest Patches | Score | Non-Fragmented Forest Area (FAD; %) | Score | Non-Fragmented Forest Area (FOS; %) | Score | Adjacency to a Protected Area | Score | Overlapping with Other Species' Distribution Area | Score | Overlapping with Other Species' Conservation Proposal | Score | Ranking |
|---|---|---|---|---|---|---|---|---|---|---|---|---|---|---|---|---|---|---|
| *Eulemur cinereiceps* | Proposal 1 | **6243.3** | **3** | **75.7** | **3** | 86.3 | **3** | 35.2 | **3** | 87.0 | 0 | Yes | 1 | 1 | 1 | 0 | 0 | **14** |
| | Proposal 2 | 648.4 | 2 | 50.9 | 1 | 61.1 | 2 | 0.0 | 0 | 0.0 | 0 | Yes | 1 | 0 | 0 | 0 | 0 | 6 |
| | Proposal 3 | 225.8 | 1 | 56.1 | 2 | 40.6 | 1 | 0.0 | 0 | 31.6 | 2 | Yes | 1 | 1 | 1 | 0 | 0 | 8 |
| *Eulemur flavifrons* | Proposal 1 | **7652.6** | **3** | **66.6** | **3** | 95.2 | **3** | 32.6 | **3** | 96.1 | 3 | Yes | 1 | 0 | 0 | 0 | 0 | **16** |
| | Proposal 2 | 7175.3 | 2 | 57.6 | 1 | 88.1 | 2 | 0.5 | 1 | 2.8 | 1 | No | 0 | 0 | 0 | 0 | 0 | 7 |
| | Proposal 3 | 2642.6 | 1 | 58.4 | 2 | 87.5 | 1 | 2.5 | 2 | 91.5 | 2 | No | 0 | 0 | 0 | 0 | 0 | 8 |
| *Eulemur mongoz* | Proposal 1 | **1645.1** | **3** | 43.9 | 1 | 57.7 | **3** | 0.0 | 0 | 0.0 | 0 | Yes | 1 | 0 | 0 | 0 | 0 | 8 |
| | Proposal 2 | 315.4 | 2 | 62.6 | 2 | 57.4 | 1 | **87.5** | **3** | 82.1 | 2 | Yes | 1 | 0 | 0 | 0 | 0 | 11 |
| | Proposal 3 | 269.4 | 1 | **70.4** | **3** | 94.8 | 2 | 10.6 | 2 | **98.4** | **3** | Yes | 1 | 0 | 0 | 0 | 0 | **12** |
| *Hapalemur alaotrensis* | Proposal 1 | 545.1 | 1 | 15.8 | 1 | 69.1 | 1 | 0.0 | 0 | 0.0 | 0 | Yes | 1 | 0 | 0 | 0 | 0 | 4 |
| *Hapalemur aureus* | Proposal 1 | **736.4** | **3** | **80.0** | **3** | 92.9 | **3** | 24.2 | **3** | 93.3 | **3** | Yes | 1 | 1 | 1 | 0 | 0 | **17** |
| | Proposal 2 | 507.9 | 2 | 53.6 | 1 | 69.3 | 1 | 0.2 | 1 | 22.3 | 1 | Yes | 1 | 1 | 1 | 0 | 0 | 8 |
| | Proposal 3 | 325.1 | 1 | 66.4 | 2 | 73.6 | 2 | 1.3 | 2 | 74.3 | 2 | Yes | 1 | 0 | 0 | 0 | 0 | 10 |
| *Prolemur simus* | Proposal 1 | **3913.5** | **3** | 91.0 | 2 | 76.8 | **3** | 68.1 | 2 | 99.7 | 2 | Yes | 1 | 1 | 1 | 0 | 0 | 14 |
| | Proposal 2 | 2208.2 | 2 | **102.6** | **3** | 99.8 | 2 | **86.0** | **3** | 100.0 | **3** | Yes | 1 | 1 | 1 | 0 | 0 | **15** |
| | Proposal 3 | 1046.5 | 1 | 82.5 | 1 | 94.1 | 1 | 52.3 | 1 | 97.3 | 1 | Yes | 1 | 1 | 1 | 0 | 0 | 7 |
| *Varecia rubra* | Proposal 1 | **7056.5** | **3** | **83.5** | **3** | 96.7 | **3** | **55.1** | **3** | 97.9 | 2 | Yes | 1 | 0 | 0 | 0 | 0 | **15** |
| | Proposal 2 | 5766.6 | 2 | 77.0 | 1 | **97.4** | 2 | 33.8 | 1 | **99.7** | **3** | Yes | 1 | 0 | 0 | 0 | 0 | 10 |
| | Proposal 3 | 5479.1 | 1 | 77.4 | 2 | 92.8 | 1 | 41.1 | 2 | 93.6 | 1 | Yes | 1 | 0 | 0 | 0 | 0 | 8 |
| *Varecia variegata* | Proposal 1 | **11,388.7** | **3** | 89.7 | 1 | 90.6 | 2 | 68.5 | 1 | 99.0 | 2 | Yes | 1 | 1 | 1 | 1 | 1 | 12 |
| | Proposal 2 | 8885.8 | 2 | **97.2** | **3** | **99.4** | **3** | **83.6** | **3** | 99.6 | **3** | Yes | 1 | 1 | 1 | 1 | 1 | **17** |
| | Proposal 3 | 4060.2 | 1 | 96.9 | 2 | 93.2 | 1 | 83.2 | 2 | 95.7 | 1 | Yes | 1 | 0 | 0 | 0 | 0 | 8 |

**Table 4.** Potential contribution of candidate sites to the conservation of the selected critically endangered species of lemurs of Madagascar.

|  | Minimal Mean Habitat Size (ha) | Mean Number of Individuals/Minimal Patch Size | New Protected Area (ha) | New Mean Protected Populations | New Mean Protected Individuals |
|---|---|---|---|---|---|
| *Eulemur cinereceips* | 14 | 8 | 6243 | 446 | 3567 |
| *Eulemur flavifrons* | 7.5 | 8 | 7653 | 1020 | 8163 |
| *Eulemur mongoz* | 3 | 3 | 269 | 90 | 269 |
| *Hapalemur alaotrensis* | 2 | 6 | 535 * | 268 | 1605 |
| *Hapalemur aureus* | 20 | 3 | 736 | 37 | 110 |
| *Prolemur simus* | 50 | 6 | 2208 | 44 | 265 |
| *Varecia variegata* | 106 | 12 | 8886 | 84 | 1006 |
| *Varecia rubra* | 41 | 18 | 7057 | 172 | 3098 |

(*) Already in a protected area.

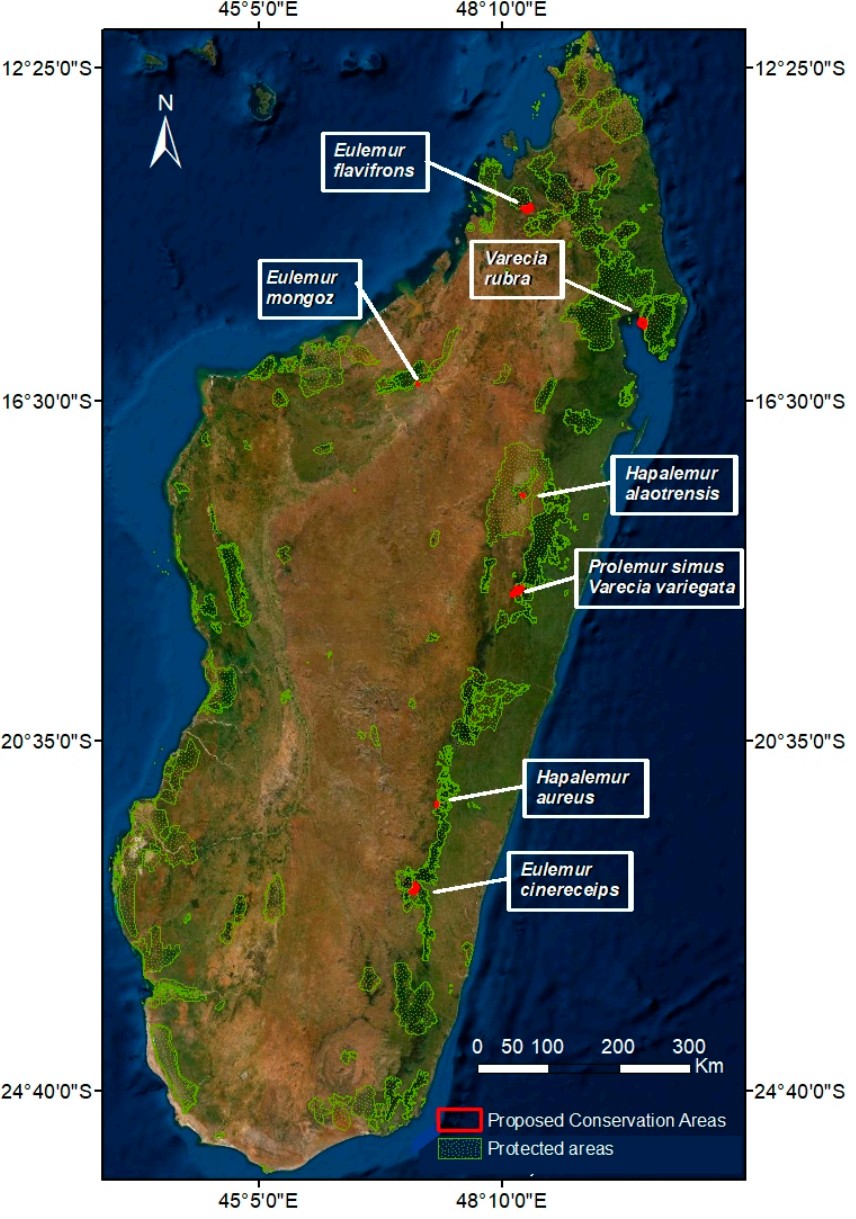

**Figure 3.** Location of the proposed new conservation areas (red) and their relative position to protected areas (green) according to the World Database on Protected Areas. Source: Esri, Maxar, GeoEye, Earthstar Geographics, CNES/Airbus DS, USDA, USGS, AeroGRID, IGN, and the GIS User Community.

**Table 5.** Existing protected areas adjacent to the final candidates to become new conservation sites of critically endangered lemur species.

| Lemur Species | Adjacent Protected Area | Legal Category | IUCN Category |
|---|---|---|---|
| *Eulemur cinereceips* | Andringitra | National Park | II |
| | Corridor Forestier | Protected Landscape | V |
| | Ambositra Vondrozo | | |
| *Eulemur flavifrons* | Manongarivo | Special Reserve | IV |
| *Eulemur mongoz* | Ankarafantsika | National Park | II |
| *Hapalemur alaotrensis* | Lac Alaotra | Ramsar site | V |
| *Hapalemur aureus* | Corridor Forestier | Protected Landscape | V |
| | Ambositra Vondrozo | National Park | II |
| | Ranomafana | | |
| *Prolemur simus* | Corridor Ankeniheny | Protected Landscape | V |
| | Zahamena | | |
| *Varecia variegata* | Mantadia | National Park | II |
| | Torotorofotsy | Ramsar site | V |
| *Varecia rubra* | Masoala | National Park | II |

## 4. Discussion

### 4.1. New Conservation Sites

The seven new candidate sites would protect the rapidly declining, critical habitat for a substantial number of populations of some of the critically endangered lemur species of the country while additionally taking Madagascar slightly closer to the 30% protection by 2030 target [42], improving forest connectivity in a highly fragmented landscape [43] and contributing to SDG 15 [44].

Similar to other natural areas in developing tropical countries, loss and fragmentation of natural habitats are widespread issues across the Malagasy landscape, even inside PAs [13,45]. Even though small remnants of natural ecosystems, as spatially limited as individual natural trees, can provide valuable refuge for native endangered species [46–48], species' richness declines and extinction risk increases with diminishing natural area [34,45,49]. Decreasing suitable habitats poses a serious threat to numerous endemic species restricted to one or few sites, much more so if such sites are surrounded by unsuitable habitat for dispersal and colonisation [43]. Actually, landscape connectivity plays an essential role in the mid-term conservation of forest-dependent species, such as the ones studied here, providing opportunities for individual dispersal and necessary genetic exchange among isolated populations [43]. The current Malagasy PA system includes a number of ecological corridors designated under IUCN category V [50]. However, this lenient legal regime is likely to be insufficient to warrant effective landscape-scale conservation in the face of current pressures [13,45]. Moreover, ecological connectivity is not stated as one of the objectives in the country's law on PAs [51]. Lack of explicit consideration of PAs as nodes of functional ecological networks across the landscape can be considered a substantial regulatory weakness given the quick deforestation pace on the island, the foreseeable negative effects of climate change on native biodiversity and the subsequent rising isolation of wild populations in PAs.

Targeted expansion of PA systems can result in the optimal representation of priority taxa at minimised costs in terms of resources needed and additional conservation area [52]. Some previous studies proposed priority conservation areas in Madagascar for a range of endemic taxa [52] or for specific taxa such as trees [53]. Dual ecosystem–species approaches to conservation planning such as the one proposed here (forests–lemurs) have been deemed the most efficient ones [54], although the ecological role of some focal species selected for conservation planning is sometimes not sufficiently understood and, as a result, purely spatial prioritisation approaches have been questioned [55]. On the other hand, protecting flagship species such as lemurs can benefit the conservation of other less charismatic species of similar or greater ecological value that share their habitats and ecological requirements [34,56–58]. The habitats of all these species can be preserved, resulting in fewer land use changes, reduced invasive species colonisation and higher environmental

quality [59]. In addition, the protection of migrating umbrella species may encourage the designation of ecological corridors [60]. Our priority areas for conservation largely coincide, geographically, with those included in a previous multi-factor prioritisation study of 98 lemur species inside a sample of PAs of Madagascar, which highlights the importance of protected sites along the wet tropical forest of the eastern side of the country in terms of lemurs' threat level, species richness and evolutionary distinctiveness [19].

The proposed new conservation sites for *P. simus*, *V. rubra* and *V. variegata* present both large and untouched forest patches, which make them promising areas of conservation. The final new conservation areas of *P. simus* and *V. variegata* coincide, the area or the later species being larger and containing the proposed conservation area of *P. simus*. This is the optimal, most efficient situation, as the conservation efforts will benefit both species. In the case of *E. mongoz*, the area is small, but there are at least two intact forest patches. Our spatial analysis revealed that the best forest patches for the conservation of *E. cinereiceps*, *E. flavifrons* and *H. aureus* were moderately fragmented, for which a double action designating sites as PAs and, at the same time, internally restoring the proposed forest patches would be advisable. Ecological restoration followed, where needed, by adequate protection could substantially help to restore functional landscapes and reduce the extinction risk of many isolated, endangered species in Madagascar [43].

The case of *H. alaotrensis* is singular as it is not strictly a forest species, but rather a pristine wetland specialist species [32], so its proposed forest patches are small and dispersed, which might not be as alarming as for the rest of forest species. Thus, it was not possible to propose three potential new conservation areas for a final assessment, but only a small one of 550 ha, with barely 85 ha of forests. Given *H. alaotrensis'* very limited distribution area, high dependency on wetland vegetation and extremely high risk of extinction, its conservation stands out as a conservation priority from this study. Combining rapid, increased protection of its remaining habitat with providing a new suitable habitat for the species through ecological restoration seems the safest way to ensure the short-term survival of this species [53,61]. Aware that conservation efforts need to be complemented by restoration of degraded ecosystems [62], the international community has deemed 2021–2030 the Decade on Ecosystem Restoration, 'with the aim of supporting and scaling up efforts to prevent, halt and reverse the degradation of ecosystems worldwide and raise awareness of the importance of successful ecosystem restoration' [63]. The extent of forest ecosystem degradation in Madagascar makes ecological restoration a useful tool for increasing ecosystems' integrity and species' chances of survival in the wild. However, the high cost of active restoration activities and the limited availability of funds among Malagasy stakeholders make it advisable to carefully select the most appropriate sites where restoration is likely to be most impactful at the lowest possible cost [61,64].

Given that all candidate sites are on the border of existing PAs or inside PAs (in the case of *Hapalemur alaotrensis*) the proposed protected sites could most efficiently be included in existing PAs by enlarging the PAs' borders, thus making the most of existing managerial resources and networks, reducing costs and facilitating engagement with local populations [65,66]. *Hapalemur alaotrensis* is, again, a singular case, with a very small distribution area (of roughly 21,000 ha) completely included in the Lac Alaotra Ramsar site. Ramsar sites have been shown to provide uneven protection for biodiversity, even when management plans are in place [67]. In Africa, Ramsar sites were reported ineffective for the protection of waterbirds in the face of intense human resource use [68]. Thus, the candidate site for *H. alaotrensis* (as well as most of its current distribution area) would ideally take the form of a new PA with greater legal stringency and potential effectiveness such as a nature reserve or national park (IUCN category I or II) inside the current Ramsar site [50].

It is important that local populations have their say in PA management, benefit from nature conservation and are aware of such benefits to increase their engagement, buy-in and regulation compliance, thus facilitating management and conservation [69]. The ecosystem service and economic potential of conserving forest areas must be clearly understood so that local communities realise the serious mid-term negative impacts of their customary

slash-and-burn practices not only for biodiversity, but also for them in terms of soil erosion and declining agricultural outputs [70]. Lemurs are not just irreplaceable biological and evolutionary conservation units but also fundamental tourism assets in the country [6]. The current socioeconomic scheme in place in Malagasy PAs that entails income sharing from entrance fees with local communities, earned from a dual national–foreigner pricing policy that includes entrance fees to PAs plus compulsory guide hiring (most of them locals), is likely to offset opportunity costs and enhance the living standards of local populations through nature-tourism-related activities. Other types of PA-related jobs, such as ranger, greatly benefit from local knowledge, networks and influence while at the same time enhancing the wellbeing of local residents, thus countering negative incentives and trends for destroying remaining forests [19].

### 4.2. Methodological Remarks

Size, shape and fragmentation are related to the conservation potential of habitats [34]. The methods used in this study allowed both a qualitative and a quantitative assessment of the suitability of forests to host the studied lemur species. GUIDOS is based on a methodology used by the USDA Forest Service [71,72] and the Montréal Process [73] for the assessment of forest ecosystems, which was adapted by the European Commission for the evaluation of European habitats [40,41]. The total size of the proposed conservation areas and the total forest area within them were not always the definitive criteria to select an area. In the case of *E. mongoz*, *P. simus* and *V. variegata*, the selected areas were significantly smaller than the largest proposal. In the case of *E. flavifrons*, the final candidate was eight times smaller than the largest proposal. However, the forest within them was less fragmented and further away from human impacts and therefore, of greater conservation potential.

Many other variables could have been taken into consideration in the study for the selection of the most suitable areas for conservation. From internal communications with the University of Antananarivo, the mining and wood logging lobbies were pointed out as important factors in the final designation of new PAs. However, the location and GIS data for these are confidential and could not be used in this study. Nevertheless, remote sensing and GIS are useful tools to scout potential new conservation areas on a broad scale, especially considering the large territory of Madagascar and the limitations to exploring some remote forest areas in the country [74]. Our recommendations can serve as a starting point for nature conservation managers and decision-makers to take urgently needed decisions about the conservation of the studied lemur species. Additionally, this work must be complemented by field assessments that verify the rapidly changing environmental and socioeconomic conditions of each site on the ground and consider other factors, such as the confluence of other economic activities or small human settlements, which may influence the decision of setting an area aside for nature conservation [52].

### 5. Conclusions

Protecting, expanding and connecting the rapidly waning natural forests of Madagascar and other biodiversity-rich tropical areas is a worldwide conservation priority. Seven sites stood out as priority forest sites for the conservation of the eight lemur species selected here. They were the best conserved and least pressured sites according to the criteria used. Protection of those sites could add substantial new, protected quality habitat for the critically endangered lemur species studied here in a socioeconomically feasible and efficient manner, as all such sites are adjacent to or inside (for *H. alaotrensis*) existing PAs. In the case of *H. alaotrensis*, the legal upgrading of its small remaining habitat, together with the expansion of its distribution area through ecological restoration of wetland area, where possible, emerged as an urgent conservation action. The fact that some of the country's major PA-managing organisations are involved in this study increases the chances of its results being used to improve the conservation of the selected lemur species and other species of conservation interest living in this priceless land for global biodiversity.

**Author Contributions:** Conceptualization, D.R.-R. and V.E.G.M.; Methodology, V.E.G.M. and D.R.-R.; Software, V.E.G.M.; Formal analysis, V.E.G.M.; Investigation, V.E.G.M. and D.R.-R.; Data curation, V.E.G.M., G.M.-G. and A.M.O.; Writing—original draft preparation, D.R.-R. and V.E.G.M.; Writing—review and editing, V.E.G.M., D.R.-R., G.M.-G., A.A., L.O.R., A.B. and D.A.M.; Project administration, D.R.-R.; Funding acquisition, D.A.M. All authors have read and agreed to the published version of the manuscript.

**Funding:** This research received no external funding.

**Institutional Review Board Statement:** Not applicable.

**Data Availability Statement:** Species distribution areas (IUCN, 2021 [18]); World Database on Protected Areas (UNEP-WCMC [37] & IUCN, 2021 [18]); pen Street Map (OSM, 2021 [38]); land cover map of Madagascar (Zhang et al., 2020 [36]).

**Acknowledgments:** We would like to acknowledge the useful remarks by three anonymous reviewers and editors, which helped to enhance the quality of our manuscript. This study recognises contributions through the sequence-determines-credit approach.

**Conflicts of Interest:** The authors declare no conflict of interest.

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
