# Peer review of "Identification of Priority Forest Conservation Areas for Critically Endangered Lemur Species of Madagascar"

_land, doi:10.3390/land11091455_

Round 1

Reviewer 1 Report

Review report:

   This study uses spatial analysis combing several spatial and ecological criteria to identify the prioritized forest conservation areas for the critically endangered lemur species in Madagascar. The proposed methodology in this study is sound and piratical for the study area. This study has practical implications for the conservation of the critically endangered lemurs in Madagascar and is suitable for the scope of the journal Land. Therefore, I suggest accepting this study for publication consideration.

For the methodology design:

     The previous study shows that water availability is necessary for to lemur survive. I suggest taking into account the water availability in the methodology design,e.g. adding the distance to the river. In addition, the climate is also an essential factor for evaluating the lemur conservation sites. It would be great to quantify the future climate risk of the currently identified conservation areas in this study. This will increase lemur reserve priority.

Minor comments:

1.    In table 2, the number format is inconsistent. It should be 1.1, 1.2, 1.3, 1.4, and 1.5. Please check it.

2.    In the methodology design, the distance of human settlements is set at 1 km. I think this distance could be increased (e.g., 5 km) to decrease the pressure from human activities.

3.    It isn't easy to see the legend of Figure 1, suggesting improving the clarity of figure 1.

Author Response

  1. The reviewer makes a sensible point suggesting including watercourses and masses in the analysis. Nevertheless, lemur species are well adapted to scarce water availability. In dry areas of the country (largely outside the proposed areas), lemurs have been seen drinking from empty shells in the forest, trunk holes, flowers and leaves. They also get an important part of their drinking requirements from their vegetal diet, which they adapt in terms of water content according to the season[1], thus not making them so dependent on surface water availability. Moreover, all the identified priority areas were located in the north and eastern parts of the country, where rainfall is the greatest and surface water availability is too.

  1. The suggestion to quantity the risk of climate change in the identified priority areas is certainly relevant in a globally changing context, even though climate change is an ancillary threat to lemur conservation, far behind from the main one: deforestation, which was the objective of our study. Notwithstanding the interest of considering climate change as a follow-up study, such new substantial analysis would deserve a study on its own and is thus outside the means and scope of the current study.

  1. Numbering in Table 2 was corrected, as suggested.

  1. We agree that 1km may not be a very big distance to reduce pressures from human settlements. We had tried different distances, starting from 10km and under. However, the largely dispersed small settlements in rural Madagascar cancelled almost all the available area when applying larger buffers and became a very restrictive criterion for the selection of suitable areas (the same happened for other pressures, notably croplands). Thus, we were forced to make some spatially feasible proposal by reducing the buffer distances to pressures. In its current form, at least we made sure that priority areas are not adjacent to important pressures, which makes our proposal both technically and practically feasible.

  1. Figure 1 was improved in its visual quality and included in a country-wide geographical context, as required.

[1] https://link.springer.com/article/10.1007/s10329-013-0392-0

https://ielc.libguides.com/sdzg/factsheets/ringtailedlemur/diet

Reviewer 2 Report

The article is interesting and could make an important contribution to the field, but unfortunately in its current form the manuscript lacks research depth, visible by a focus on the case study rather than the research issue, proved by poor discussions and conclusions. Thus, the manuscript requires a strong development of these sections. Detailed comments are provided for each section of manuscript.
The research goals (lines 86-88) should be placed in a separate paragraph, to better emphasize them. Also, based on the shortcomins of the previous studies (line 85), it would help to add after the research goals 1-2 sentences showing how these goals are addressing the previous shortcomins, and what the novelty of research is, compared to previous studies.
Figure 1 should present the study area location in an broader context, making visible the entire map of Madagascar, so that a Brazilian researcher could understand it too, perhaps by using the map-in-map system.
The most important section of a research article, the Discussions, is insufficiently developed. The section is meant to emphasize the importance of research, justifying its publication. Normally, this section includes include (A) the significance of results - what do they say, in scientific terms; (B) the inner validation of results, against the study goals or hypotheses; (C) the external validation of results, against those of similar studies from other countries, identified in the literature; (D) the importance of the results, meaning their contribution (conceptual or methodological) to the theoretical advancement of the field; (E) a summary of the study limitations and directions for overcoming them in the future research. Out of these, only a discussion of the significance of results and methodological details is present. The "Discussions" should be developed to include the missing elements.
Conclusions are not sufficiently broad in scope, and lack research depth, pertaining only to the case study and being in fact just a summary of the main findings. Conclusions are meant to deliver a scientific message, far away beyond the case study, to the entire scientific community, making a clear contribution to the theoretical (conceptual or methodological) development of the field. Conclusions must be developed.

Author Response

  1. The reviewer is right that the Discussion may have been too narrowly focused on the case studies that we found. Thus, we notably expanded it and framed it covering broader scientific topics, adding a number of new relevant references for better context and understanding.

  1. We placed the study’s objectives in a separate paragraph and provided some explanations why it helps overcome some existing conservation planning issues in Madagascar, as suggested.

  1. We agree that Figure 1 was out of its geographical context, so we put in the country’s context for better understanding, as suggested.

  1. The Conclusions were also expanded to cover the topic more broadly, as required.

Reviewer 3 Report

Revision

This is a very interesting and relatively well written manuscript on an area of very high conservation concern (with a large number of dramatically threatened species). I am not a Mother Tongue in English: however language and style are apparently good. Logic is clear. Procedure of selection is useful for conservation managers and policies. Therefore I think that this ms deserves to be published after MINOR REVISIONS, as soon as possible to help conservation strategies for these rare species. I suggest only some minor (but important) points about the role of corridors (only one citation in the text!) and ecological networks as a tool to connect nature reserves (see Bennett, A.F. (1999), Linkages in the Landscapes. The Role of Corridors and Connectivity in Wildlife Conservation, Gland, Switzerland, and Cambridge: IUCN). These considerations could be also added in conclusions (for example, as recommendations or ‘implications for conservation’).

MAIN POINTS

I think that there is only a point of weakness in this manuscript: the role of corridors, connectivity and ecological network policies to improve populations of these mammals. Also 1-3 sentences at the end of the manuscript could be added. Insularization of nature reserves and forest patches) may represent a serious threat for the maintaining of viable populations of lemurs. Please add some considerations. I suggest some references in this regard. For example, there is a large amount of considerations on the role of ecological networks as a tool to connect nature reserves in fragmented landscapes. see : Journal of Land Use Science8(2), 215-223, 2013. Available at: https://www.tandfonline.com/doi/full/10.1080/1747423X.2011.639098.

About site selection there are also seminal papers that I suggest (for example: ‘Protected area surface extension in Madagascar: Do endemism and threatened species remain useful criteria for site selection ?’ published on Madagascar Conservation & Development5(1).: 35-47. Available at: https://www.ajol.info/index.php/mcd/article/view/57338. See also: https://www.taylorfrancis.com/chapters/edit/10.4324/9780203118313-21/conservation-politics-madagascar-expansion-protected-areas-catherine-corson

MINOR POINTS

Row 1: there is an ‘8’ before ‘Article.

Row 44. ‘alaotrensis’ in italic.

Improve the readability of the Fig. 1. It is better in high definition.

Results: rows 185-188. ‘The quality of a potential new conservation area for lemurs was based on the quality of their forests from a structural point of view, considering factors such as total forest area, 186 average patch size, number of forest patches, spatial aggregation and/or dispersion of forest patches, compactness, amount of forest edges, amount of forest perforations, and inter-patch connectivity.’: this sentence is methodological. Why this sentence is at the beginning of Results?

Row 192, 1° column of Table 3, Figure 3 and everywhere. Scientific names should be written in italic font.

Add the role anonymous reviewers and Editors i the Acknoweledgments.

Have a nice work.

Author Response

  1. The reviewer is right that forest fragmentation was insufficiently discussed in the original version. Thus, we notably expanded that topic in the Discussion and Conclusions, adding a number of new relevant reference, as suggested. We thank the reviewer for the literature suggestions.

  1. We deleted the “8” from row one, as suggested.

  1. We changed alaotrensis to italics in row 44 and elsewhere across the text, as suggested.

  1. We improved the visual quality of Fig. 1 and included it in a countrywide geographical context for easier understanding.

  1. The paragraphs between lines 185-188 and 211-215 were indeed methodological and were moved to the Methods. The corresponding section of Results was rearranged accordingly.

  1. The reviewers and editors were acknowledged for their contribution to the article’s quality, as suggested.

Round 2

Reviewer 2 Report

The authors have fully and deeply addressed my previous comments, and as a result the manuscript increased its research depth and addresses a broader international audience. I do not have any additional comments and recommend its publication in the revised form.